# High-Density EEG in a Charles Bonnet Syndrome Patient during and without Visual Hallucinations: A Case-Report Study

**DOI:** 10.3390/cells10081991

**Published:** 2021-08-05

**Authors:** Andrea Piarulli, Jitka Annen, Ron Kupers, Steven Laureys, Charlotte Martial

**Affiliations:** 1Department of Surgical, Medical, Molecular Pathology and Critical Care Medicine, University of Pisa, 56126 Pisa, Italy; andrea.piarulli@uliege.be; 2Coma Science Group, GIGA-Consciousness, University of Liège, 4000 Liège, Belgium; jitka.annen@uliege.be (J.A.); steven.laureys@uliege.be (S.L.); 3Centre du Cerveau, University Hospital of Liège, 4000 Liège, Belgium; 4Department of Neuroscience, University of Copenhagen, 1050 Copenhagen, Denmark; endogonidia@gmail.com

**Keywords:** Charles Bonnet syndrome, EEG, visual hallucination, resting state

## Abstract

Charles Bonnet syndrome (CBS) is a rare clinical condition characterized by complex visual hallucinations in people with loss of vision. So far, the neurobiological mechanisms underlying the hallucinations remain elusive. This case-report study aims at investigating electrical activity changes in a CBS patient during visual hallucinations, as compared to a resting-state period (without hallucinations). Prior to the EEG, the patient underwent neuropsychological, ophthalmologic, and neurological examinations. Spectral and connectivity, graph analyses and signal diversity were applied to high-density EEG data. Visual hallucinations (as compared to resting-state) were characterized by a significant reduction of power in the frontal areas, paralleled by an increase in the midline posterior regions in delta and theta bands and by an increase of alpha power in the occipital and midline posterior regions. We next observed a reduction of theta connectivity in the frontal and right posterior areas, which at a network level was complemented by a disruption of small-worldness (lower local and global efficiency) and by an increase of network modularity. Finally, we found a higher signal complexity especially when considering the frontal areas in the alpha band. The emergence of hallucinations may stem from these changes in the visual cortex and in core cortical regions encompassing both the default mode and the fronto-parietal attentional networks.

## 1. Introduction

Charles Bonnet syndrome (CBS) is a rare syndrome characterized by the appearance of recurrent and complex visual hallucinations in elderly people without mental disorders, who generally recognize their hallucinations as unreal [1]. The syndrome is typically reported by people who have a severe decrease or complete loss of vision, often caused by retinitis pigmentosa, cataracts, macular degeneration, or optic neuritis [2].

Though CBS was first described in the 18th century by the biologist who gave his name to the syndrome [3], the neurobiological mechanisms underlying the hallucinations remain elusive, with no satisfactory response as to why some blind patients hallucinate and many do not. Nonetheless, some attempts have been made to find plausible explanations. Currently, research has shown that CBS may be associated with lesions in the visual system, ranging from the retina to the occipital cortex [4,5,6,7,8,9], likely causing the visual hallucinations. So far, the most widely accepted mechanism of the emergence of such visual hallucinations refers to the “release phenomenon” suggesting that deafferentation of the visual cortex might lead to cortical hyperexcitability in this area [10,11]. While a few studies, including electroencephalography (EEG) ones, suggest that hyperexcitability in the visual cortex of partially blind CBS patients is associated with external visual stimuli (e.g., [12,13]), other studies demonstrate that CBS visual hallucinations can rather arise in the absence of external visual stimuli and, thus, suggest the possibility that spontaneous brain activity could generate conscious percepts (e.g., [14]). Since CBS is relatively rare [15], its investigation is only in the preliminary stages. Only a few studies have investigated the electrophysiological correlates in CBS and none of these investigations has ever used high-density EEG.

We here describe the case of an 85-year-old man with late-onset visual impairment and CBS. First, the patient underwent neuropsychological, ophthalmologic, and neurological examinations. Next, high-density EEG was performed as long as necessary to record data during rest and while vivid visual hallucinations were present. Investigating and comparing both conditions may provide novel insights into the neural pathways that underlie the emergence of visual hallucinations in CBS.

## 2. Materials and Methods

### 2.1. Case Report

An 85-year-old man with visual impairment and no psychiatric history presented to the Centre du Cerveau² (University Hospital of Liège, Belgium). The patient suffered from retinitis pigmentosa, a degenerative eye disease characterized by progressive degeneration of the rod photoreceptors in the retina, which typically leads to severe vision impairments [16]. He was three when he began experiencing a progressive loss of vision. Some years later, during the teenage years, he experienced a progressive development of “tunnel vision”, i.e., his peripheral field of vision progressively narrowed. Concurrently, the patient developed hemeralopia, a night vision deterioration caused by the abolition of rod cells. At the age of 70 years, he lost his central vision, causing complete blindness at the age of 80 years. He also reported a positive family history of CBS.

The patient described a five-year history of increasing frequency of visual hallucinations. His visual hallucinatory experiences started at the age of 80, together with the occurrence of his complete blindness. Overall, he was able to report a coherent and very detailed description of his hallucinations, as well as its occurrence context. The patient reported that his visual hallucinations gradually became more frequent and could occur many times during the day. He described his hallucinations as, in general, well formed, ranging from simple flashes or colored backgrounds to more complex with the appearance of faces, objects, people, or landscapes. They were usually binocular, covering the entire visual field and could vary in size and color. He nevertheless specified that animations (i.e., scenes in motion) were only present in his right visual field. The visual hallucinations generally occurred when he has his eyes open, and they did not disappear when he closed his eyes. They were never accompanied by abnormal perception in any other sensory modality. He reported his hallucinations as pleasant. The patient is apparently not able to consciously control the occurrence and content of the visual hallucinations. He was fully aware of their unreal nature.

### 2.2. Procedure

The patient went through a consultation with a neurologist, as well as with a neuropsychologist. The latter examination included the administration of the Mattis Dementia Rating Scale [17] and of the Montreal Cognitive Assessment (MoCA, [18]) using the version adapted for blind individuals (MoCA-BLIND, [19]). Given his visual impairments, these cognitive tests were administered verbally, thereby omitting all vision-specific items. The patient was then submitted to an ophthalmologic examination, notably including measurements of visual acuity and visually evoked potentials.

The patient finally underwent a high-density EEG recording session. The EEG was recorded using a Net Amps 300 system (GES300) with a 256-electrode Hydrocel Geodesic Sensor Net (Electrical Geodesic Inc., Eugene, OR, USA). EEG channels were referenced to the vertex, electrode impedances were kept below 50 kΩ throughout the recording, and signals were acquired at a sampling rate of 250 Hz, using Electrical Geodesic Net Station software version 4.5.4 (Electrical Geodesic Inc., Eugene, OR, USA). During the EEG acquisition, which lasted about an hour, the patient was comfortably seated in a quiet room at the hospital. During the first 17 min, the patient was in an eyes-closed resting state condition, thus providing a baseline measure of brain electrical activity. After the resting state period, he started having vivid visual hallucinations, which lasted for about 20 min. During the hallucinatory period, he described the visual hallucinations he was experiencing, giving additional details about the hallucinations at the end of the EEG acquisition.

An MRI exam was also performed; the results are reported in [20].

### 2.3. EEG Pre-Processing

EEG recordings were analyzed using tailored codes written in MATLAB (MathWorks, Natick, MA, USA) and taking advantage of EEGLAB toolbox functions [21].

We analyzed about 37 min of the one hour recording due to the presence of massive movement and muscular artifacts especially during the second part of the recording (visual hallucination (VH) period): 17 min in the resting state (RS) condition, and 20 min in the VH condition were retained for further analyses, thus discarding the last 23 min. Signals were filtered between 0.5 and 30 Hz. Channels on the forehead, neck, and cheeks, which mostly contribute to movement-related noise were discarded [22], thus, retaining 183 channels out of 256. For each condition (RS and VH conditions), epochs with signals exceeding 100 μV were automatically discarded. Retained signals were visually inspected for the removal of large non-stationary artifacts and noisy channels. The retained epochs of each phase were then concatenated and submitted to independent component analysis (ICA) to remove ocular and/or muscular artifacts [23]. Noisy signals were then substituted with signals obtained via spline-interpolation [24], and signals were average-referenced. At the end of the artifact’s rejection procedure, 7.5 min of the RS condition and 8 min of the VH condition, were kept for further analyses. Even after the artifact removal procedure, the EEG visual inspection highlighted the presence of residual high levels of electromyographic noise caused by involuntary muscle movements both during RS and VS periods. On this basis, we restricted our analyses to delta (1–4 Hz), theta (4–8 Hz), and alpha (8–13 Hz) bands, where the influence of EMG noise is negligible [25]. As a final pre-processing step, both RS and VS conditions were divided into 15 s non-overlapping epochs (thus obtaining 30 and 32 epochs, respectively).

### 2.4. Spectral and Connectivity Analyses

A spectral analysis was conducted for each condition. The EEG power spectral density (PSD) was obtained for the bands of interest: delta (1–4 Hz), theta (4–8 Hz), and alpha (8–13 Hz). Power density distributions were estimated by applying a Hamming-windowed FFT on 15 s consecutive epochs. For each epoch and electrode, the mean band PSD was estimated as the average over its frequency bins. For each condition, the PSD of each band and electrode was finally obtained by averaging among epochs and then log-transformed.

Both for RS and VH, we estimated the connectivity between each couple of electrodes using the debiased weighted phase lag index (dwPLI, [26]). The dwPLI at frequency bins of 0.5 Hz was estimated for each 15 s epoch (for the purpose, each epoch was segmented into 2 s periods with a 50% overlap between contiguous ones). For each epoch and couple of electrodes, the dwPLI in each band of interest was estimated by averaging over its frequency bins. The dwPLI of each condition, band, epoch, and couple of electrodes was finally calculated by averaging over the time samples belonging to the epoch itself. For each condition and epoch, the dwPLI values across all channel pairs were used to construct 183 × 183 connectivity matrices for delta, theta, and alpha bands. Scalp power maps were visualized using EEGLAB functions [21], while connectivity maps were displayed taking advantage of the Brain Net toolbox [27].

### 2.5. Graph Analysis

Connectivity matrices were thresholded by varying the connection density to retain between 90% and 10% of the higher connectivity values in steps of 2.5% [28]. At each density, the matrix was represented as a weighted graph, with channels as nodes and non-zero connectivities as links between the nodes. Each graph was characterized by using graph-theoretical parameters implemented in the Brain Connectivity Toolbox [29]. We adopted the following of metrics:Strength: The strength of a node is defined as the sum of its edges. The mean graph strength is, thus, estimated as the average over nodal strengths.Local efficiency provides a measure of the degree of information integration between the immediate neighbors of a given network node. The mean local efficiency thus reflects the degree of local connectivity within a graph [30].Global efficiency provides a measure of network integration and is defined as the average inverse shortest path length [30].Modular structure and modularity: The modular structure of a graph is estimated by subdividing the network into groups of nodes (maximizing the number of within-group links and minimizing the number of between-group links). Modularity indicates the degree of reliability of a given modular structure [31].Participation coefficient provides an estimate of the degree to which an included node of a given module is linked with other modules. Nodes with a high participation coefficient promote inter-modular integration and, as such, a network with a high participation coefficient is likely to also be globally interconnected.

Finally, for each condition and epoch, each metric was averaged over the considered connection densities.

### 2.6. Lempel–Ziv Complexity

For each condition, epoch and electrode, Lempel–Ziv Complexity (LZC) was estimated for each band of interest. As a first step, the EEG of both the RS and of the VH conditions were band-pass filtered (Chebyshev II filters) to obtain signals in delta, theta, and alpha bands. For each band, condition, epoch, and channel, the EEG signal was converted into a binary sequence. The threshold used to binarize the signal for each band, epoch, and electrode was computed in line with the following [32]:The average signal value (over the single epoch) was estimated and subtracted from the signal’s original time series (*s_je_*); the resulting signal was then linearly detrended (*s_je_**). Note that *j* identifies the *j*th channel and *e* the *e*th epoch.The analytic signal of *s_je_** was estimated using the Hilbert transform.The binarization threshold (*th_je_*) for each channel and epoch was obtained as the average over the epoch of the analytic signal absolute value.For each band, the EEG signal at each electrode and epoch (*s_je_*) was binarized based on the estimated threshold (*th_je_*). If *s_jek_* ≥ *th_je_*, *sb_jek_* = 1, otherwise *sb_jek_* = 0. Note that *k* identifies the *k*th time sample, and *sb* is the resulting binarized signal.

After binarizing the EEG signals related to each band, epoch, and channel, each resulting binary time series was submitted to the LZC algorithm [33]. At the end of the procedure, we obtained for each band, condition, and electrode a series of LZC values (one for each 15 s epoch).

### 2.7. Statistical Procedures

#### 2.7.1. Spectral Power, Lempel–Ziv Complexity and Classical Connectivity Analyses

For each band and electrode (or couples of electrodes when considering connectivity), comparisons between VH (32 samples, each corresponding to a 15 s epoch) and RS conditions (30 samples) were performed using a single threshold permutation test for the maximum t-statistics (two-samples test, 5000 permutations; [34]; for details see Appendix A).

#### 2.7.2. Graph Metrics

For each band and graph parameter (graph strength, local efficiency, global efficiency, modularity, and participation coefficient), between-condition differences (VH-32 samples and RS-30 samples) were assessed using a permutation test on the t-statistic (two-samples test, [35]) based on 5000 randomizations. For each band, p-values (one for each parameter) were then adjusted for multiple testing using the false discovery rate (FDR) procedure [36]. Significance level was set at *p* = 0.05.

## 3. Results

### 3.1. Neuropsychological Examination

The Mattis Dementia Rating Scale and the MoCA-BLIND scores were 79/79 and 22/22, respectively. The patient’s cognitive and global functioning can be considered as normal.

### 3.2. Ophthalmological Examination

An ophthalmologic disease was diagnosed. The examination revealed retinal atrophy in both eyes, as well as a decompensated corneal grafting in his left eye. Flash visual evoked potentials were flat. During the examination, he could perceive light in both eyes. He underwent a cataract operation of both eyes.

### 3.3. Neurological Examination

A diagnosis of CBS was made by the neurologist, based on his clinical history and the results of the diagnostic and clinical assessments the patient underwent. The neurologist confirmed that the patient fulfilled the four diagnostic criteria for CBS: presence of (i) complex, repetitive, and persistent hallucinations and (ii) awareness that the hallucinations are not real; and absence of (iii) additional delusions and (iv) additional hallucinations in the other senses [1].

### 3.4. EEG

#### 3.4.1. Content of the Visual Hallucinations Experienced during the VH Condition

Vivid visual hallucinations were reported by the patient during the VH condition (during 20 min). First, he described the vision of a huge stone cathedral, with a ~100-m-high ceiling. The stones were terracotta. He described the hallucinations as really beautiful, joyful, and truly magic. The colors that he saw inspired joy and were reported as having a granulated texture. There was no animation in the scene. Then, the orange shade was changing and became darker to finally have a mysterious appearance. The patient reported what he saw at this moment as very powerful and strong. He also reported some lines of light. He stated that it was the first time he saw a cathedral in his hallucinations.

The visual hallucinations he had were binocular, covering the entire visual field. He reported not knowing why the visual hallucinations started to arise, he was not able to control their appearance or their visual content. They were not accompanied by abnormal perception in any other sensory modality, and he was fully aware of their unreal nature.

#### 3.4.2. EEG Analyses

Since the high-density EEG recording was extremely noisy, 7.5 min of RS and 8 min of VH were retained for the analyses. Furthermore, we analyzed only delta, theta, and alpha bands due to movement and muscular artifacts at higher frequencies, which were present also after the cleaning procedure both during RS and VH periods (see [25]).

##### Power Spectral Density

For each band and condition, we first estimated the average scalp map (see Figure 1, columns 1–2) and then for each band, we evaluated putative between-condition differences (Figure 1, column 3).

When considering the delta band, we observed a significant PSD decrease in the midline frontal areas during the VH condition as compared to the RS one. An analogous decrease, but one involving virtually the whole frontal area, was found when considering the theta band. Both for the delta and the theta band, the frontal decrease was paralleled by an increase of PSD in the midline posterior areas. The alpha band was instead characterized by an increase of PSD in occipital areas covering the whole visual cortex and extending toward midline posterior areas (for statistical details, see Appendix A).

Based on whole scalp comparisons, we selected four representative electrodes (one for each area showing significant between-condition differences, and one “control” electrode): Fz, Pz, Oz, and Cz. As apparent from Figure 2, both VH and RS conditions were characterized by a peak of theta activity in the frontal areas (midline frontal theta, see Fz subplot). In line with whole scalp comparisons, both delta and theta PSD were higher during the RS condition as compared to those in VH. The same theta peak was observed when considering the Cz electrode, although in this case, no significant PSD difference was found between VH and RS, in line with findings about whole scalp comparisons (see Figure 1).

When considering the Pz derivation, we observed a local increase of PSD encompassing the theta and alpha bands, coherently for the two conditions. Of note, PSD during VH was significantly higher than that observed during RS coherently for the three bands of interest (in line with whole scalp analyses).

Finally, regarding occipital areas (Oz subplot), we found a local increase of PSD (theta, but mostly alpha band) analogous to that observed for Pz when considering the VH condition.

##### Classical Connectivity Analysis (dwPLI)

We next verified whether the VH condition induced significant changes in connectivity as compared to RS. When considering either the delta or the alpha band, we did not observe any between-condition difference (see Appendix A). At variance with the latter bands, we found a widespread decrease of connectivity within the theta band during the VH condition (Figure 3 and Appendix A). Significant decreases involved the whole frontal area and posterior-central areas of the right hemisphere.

##### Graph Theoretic Metrics

For each subject, condition, and epoch, connectivity values across all channel pairs were used to construct symmetric 183 × 183 connectivity matrices for each considered band. Connectivity matrices were then thresholded by varying the connection density to retain between 90% and 10% of the highest dwPLI values (steps of 2.5%). At each connection density, we characterized the weighted network’s topological features using the following metrics: (a) network strength, (b) local efficiency, (c) global efficiency, (d) modularity, and (e) participation coefficient. Each metric was averaged across connection densities (see Figure 4 and Appendix A). The estimated metrics as a function of connection density are presented in Appendix A.

No significant difference was found for any graph metric when considering either the delta or alpha networks (see Appendix A). On the other side, we observed a disruption of small-world attributes typical of complex networks [37] during VH as compared to RS. Indeed, theta networks showed a lower network strength, paralleled by a decrease of both local and global efficiency. On the other side, the whole scalp networks showed a higher modularity complemented by a lower participation coefficient although the latter comparison was not significant (see Figure 4). Of note, findings about theta networks were consistent across connection densities (Appendix A).

##### Lempel–Ziv Complexity

We next estimated band-wise LZC. No significant between-condition difference was found either when considering delta or theta bands (see Appendix A). A significant enhancement of alpha band complexity over large scalp areas was observed during the VH period as compared to that in the RS condition (Figure 5). Significant increases were found in the midline frontal areas, posterior areas of the left hemisphere, and centro-posterior areas of the right hemisphere.

## 4. Discussion

In the present study, the electrical brain activity of an 85-year-old patient with a late-onset visual impairment and CBS was investigated by comparing two conditions: (1) resting state (RS, without any hallucinatory experience, 7.5 min of cleaned data) and (2) visual hallucinations (VH, 8 min of cleaned data).

When looking at PSD, during the RS condition, we observed the emergence of theta activity in the frontal areas, and of theta–alpha in the posterior and occipital ones (Figure 1 and Figure 2). These findings seem at odds with those observed in healthy subjects, as during relaxed wakefulness the human brain activity is characterized by a marked rhythmic electrical activity in the alpha band (8–13 Hz; [38]), which in an eyes-closed condition shows an occipital prevalence. However, the results herein described come as no surprise, as previous studies highlighted how blind people show a reduction of alpha activity over parieto-occipital areas [39], paralleled by a reduced gray matter volume in these very same areas as previously observed in this patient [20]. Moreover, the onset and duration of blindness are predictors of the decline of alpha activity in blind people [40]. At variance with resting state phase, during the VH period, we observed the reinstatement of a “pure” low-alpha activity in occipital areas, see Figure 1. Of note, the presence of alpha activity has recently been proposed as an active mechanism for visual processing [41].

In both conditions, the patient showed a topologically widespread enhancement of slow activity (delta–theta bands) paralleled by a reduced alpha activity. A cofactor contributing to the emergence of this EEG pattern could be related to aging: indeed, as observed by Ishii and colleagues [42], physiological aging is characterized by (i) a reduction in the amplitude of alpha activity (8–13 Hz), (ii) a slowing of the background activity, and (iii) a global increase of delta (1–4 Hz) and theta (4–8 Hz) power.

As compared to RS, the VH condition was characterized by PSD decreases in the midline frontal areas for delta and theta bands. Both decreases were paralleled by PSD increases in the midline posterior areas corresponding at a cortical level to posteromedial cortical regions, thus including the precuneus. This latter structure has been associated with self and visual awareness [43], and its activity seems to correlate with self-reflection processes [44,45], including mental imagery [46]. Based on these considerations, we speculate that the activity of the posteromedial cortex could play a crucial role in the patient’s awareness of perceiving visual hallucinations instead of real scenarios. During the VH condition, coherently with what always happens when the patient is hallucinating outside (as reported during the anamnesis with the neurologist), the subject was aware of the unreality of his hallucinations, suggesting a successful and preserved reality monitoring. Interestingly, the precuneus has been identified as one of the regions of interest supporting successful reality monitoring in humans [47]. In general, CBS patients do recognize the unreality of their visual hallucination [1]: this ability may be related to the fact that they were sighted people becoming blind very late in their life. The unreality of the hallucinations is experienced as pleasant by our CBS patient, but this is not the case of all CBS patients who may, in some cases, suffer from anxiety [48].

When considering the alpha band, we observed a PSD increase during the VH condition (as compared to RS), encompassing both the visual cortex and midline posterior areas. Interestingly, the patient showed increased functional connectivity (estimated by fMRI) in the same posterior midline regions, which are included both in the secondary visual and salience networks, as compared to a sample of late-blind subjects [20]. The observed functional reorganization involving regions known to be crucial for self-awareness and for the processing of visual, salient stimuli could be related to the emergence of visual hallucinations in CBS. On the other side, frontal midline theta has been associated with cognitive control [49]: indeed, the significant decrease of this activity during the VH phase could be at the basis of an individual’s inability to control either the emergence of the hallucinations or their visual content. Finally, the significant increase of alpha activity in occipital areas is plausibly related to the appearance and development of the visual hallucinations, which have been linked to the endogenous activation of large sections of the visual cortex [6]. In CBS patients, neural populations within the visual system are deprived of external inputs due to visual loss, a condition that may lead to a hyper-sensitization to spontaneous activity fluctuations, which may in turn result in the emergence of visual hallucinations [50]. We make the hypothesis that the increase of occipital alpha activity could be the neural correlate of such a hyper-sensitization.

We have also revealed a widespread decrease of connectivity within theta band including the whole frontal area and posterior-central areas of the right hemisphere during the emergence of visual hallucinations. This widespread decrease was accompanied at a network level, by a concurrent decrease of small-worldness (lower local and global efficiency) and an increase of the network’s modularity. Interestingly, similar disruptions of small-worldness (i.e., decreased local efficiency and longer path length/decreased global efficiency) are consistently reported in studies on schizophrenia, whose core symptoms are hallucinatory experiences (mainly auditory, 75% of patients) and delusions [51,52], although about 35% of patients also experience visual hallucinations [53].

While a comparison between hallucinations in two such different pathological conditions seems at least hazardous, this parallelism is motivated by the hypothesis of common neurobiological mechanisms subtending all hallucinatory experiences be they auditory or visual [53,54]. The proposed model posits that the emergence of hallucinations stems from widespread impairments in attention and/or perception and stands on evidence from resting-state studies showing impaired connectivity within the Default Mode Network (DMN) and between brain attentional networks (i.e., ventral and dorsal attention networks; [54]). This hypothesis and the “release phenomenon” hypothesis [10,11] are not mutually exclusive. Our findings may support both, considering, respectively, the decrease of connectivity in the frontal areas and the heightened alpha activity in occipital regions during visual hallucinations.

Finally, we observed a significant enhancement of alpha band diversity (LZC) over large scalp areas including midline frontal areas, centro-posterior areas of the left hemisphere, and posterior areas of the right hemisphere during the VH period as compared to RS. Interestingly, an increase of alpha diversity was observed following the administration of N, N-dimethyltryptamine (DMT), a psychedelic drug known to induce visual hallucinations [55], as compared to the administration of a placebo, and during stroboscopically induced visual hallucinations [56]. In line with Timmermann and colleagues’ work [55], the increase in signal diversity here described, can be interpreted within the “entropic brain hypothesis” framework, which posits that the quality of any conscious state depends on the system entropy [57], and as such, signal diversity measures such as LZC are reliable indices of the richness of contents of any conscious experience be it “real” or “hallucinatory”.

We have described significant electrical brain changes in the visual cortex and in core cortical regions encompassing both the default mode and fronto-parietal attentional networks, which could contribute to the occurrence of visual hallucinations in CBS. We believe that this study provides a substantial contribution to the investigation of the electrophysiological signatures of CB syndrome at large and on the emergence of visual hallucinations linked to CBS. That said, we must acknowledge that this work is not free of limitations, and thus, our results should be interpreted cautiously as this is a case report study, which limits the generalizability of the presented results. Further studies should focus on electrical brain activity changes in a higher number of patients with CBS, in comparison with control groups such as for example matched late-blind people who do not suffer from hallucinations. Nevertheless, the CBS syndrome is quite rare. Furthermore, the spatial resolution of EEG is relatively low, and therefore, we do not have compelling evidence for the involvement of specific brain regions in the hallucinations. Finally, due to the presence of muscular artifacts, we limited our analyses to slow frequency bands (i.e., delta, theta, and alpha), preventing a thorough characterization of CBS electrophysiological correlates both during the resting state and during visual hallucinations.

## 5. Conclusions

As a concluding remark we would like to stress that, as this syndrome is relatively rare, both CBS trait and state (i.e., visual hallucinations) findings herein described, provide a first and up-to-now unique electrophysiological window on CBS and visual hallucinations at large. Indeed, we believe that the CBS can provide a useful clinical model for advancing our understanding of the electrophysiological mechanisms underlying the emergence of visual hallucinations. Future high-density EEG studies including larger samples of CBS patients are needed to confirm and extend the findings of this study.

## Figures and Tables

**Figure 1 cells-10-01991-f001:**
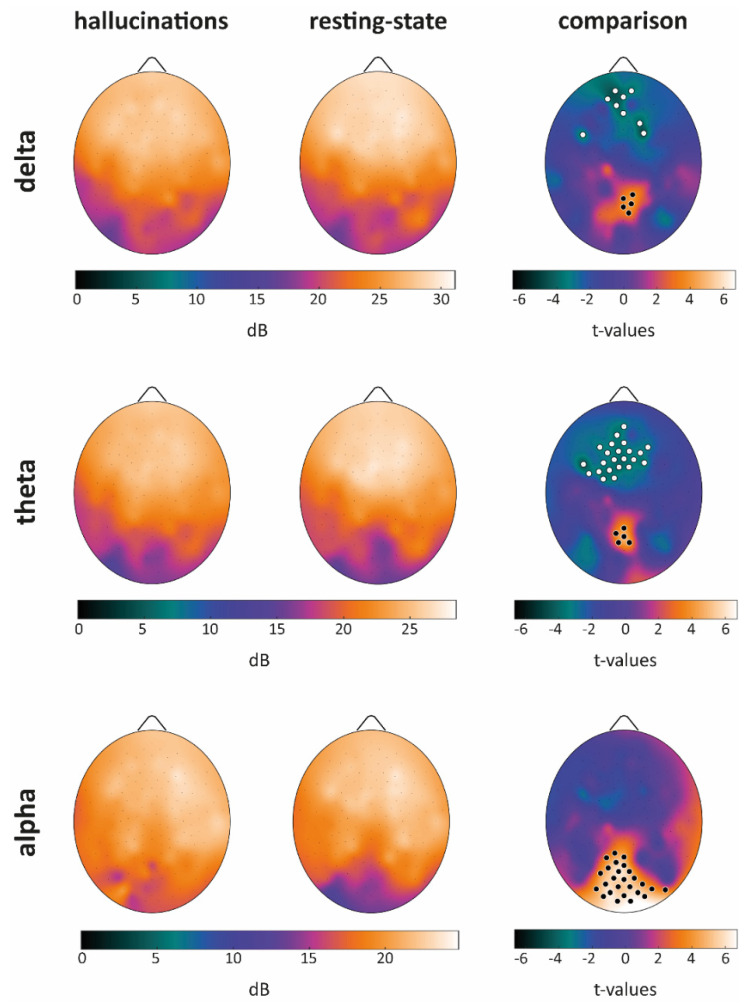
Average scalp maps for each band and condition are presented in the first two columns. Band-wise comparisons are presented in the third column. Black dots indicate electrodes showing a significantly higher PSD during the VH condition as compared to RS, white dots the opposite relationship. Critical t-values (in absolute value) for significance at *p* = 0.05 are, respectively, |t| = 3.22 for delta, |t| = 3.13 for theta and |t| = 3.12 for alpha band (single-threshold tests for the maximum t-statistics).

**Figure 2 cells-10-01991-f002:**
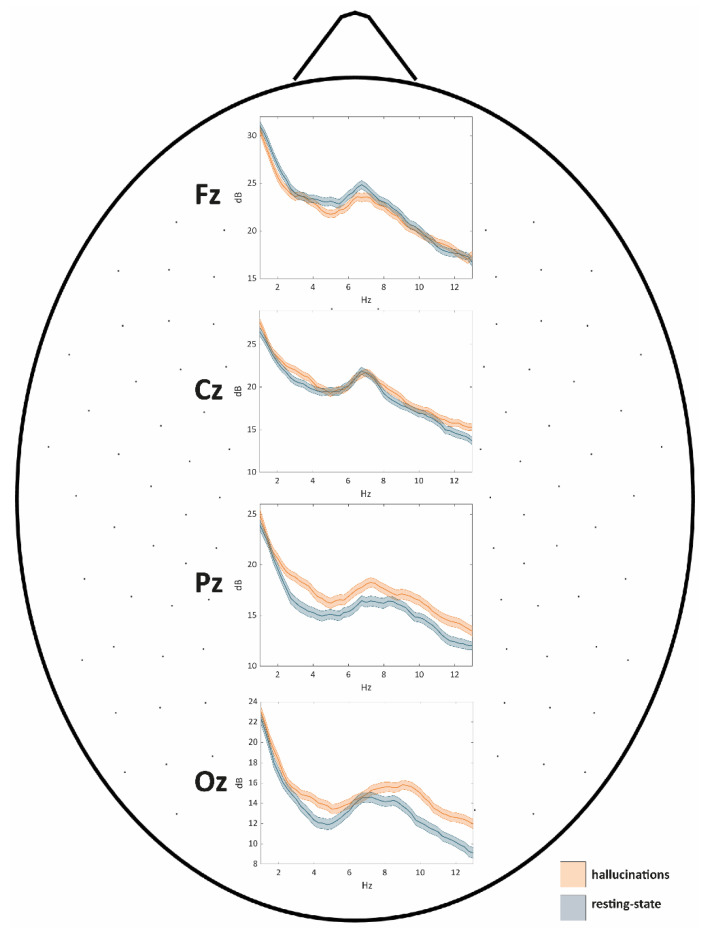
The PSD of VH and RS conditions are presented as a function of frequency for four representative electrodes. In each subplot, the solid line represents, for each condition, the mean PSD value, whereas the colored areas highlight the values enclosed between the mean-standard error and mean + standard error interval.

**Figure 3 cells-10-01991-f003:**
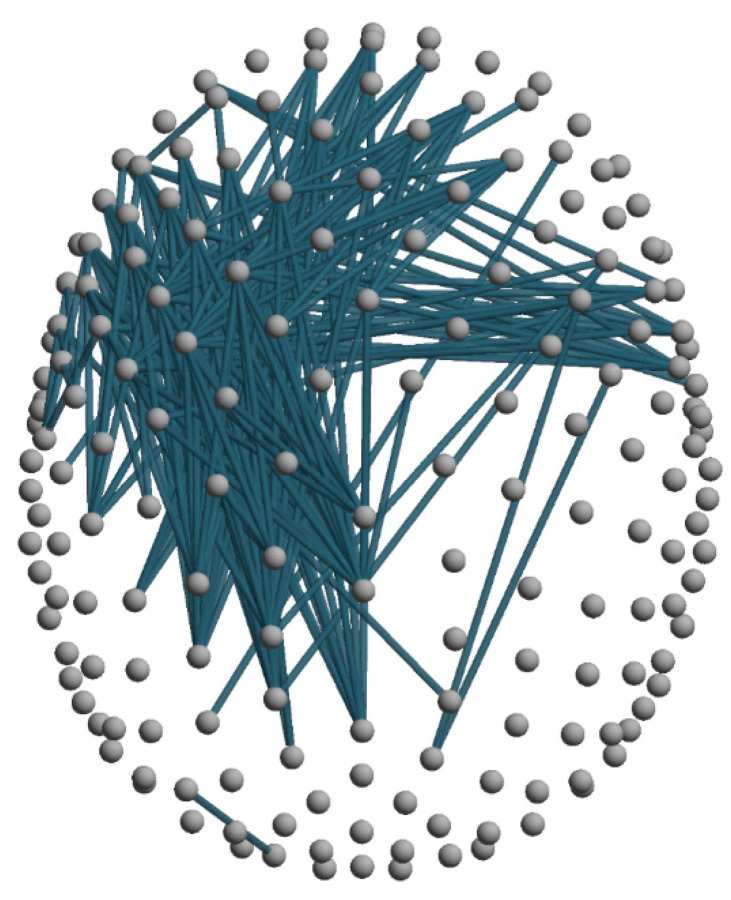
Significant decreases in theta band connectivity are presented for the VH versus RS comparisons (blue lines). The critical t-value (in absolute value) for significance at *p* = 0.05 is |t| = 3.12 (single threshold tests for the maximum t-statistics).

**Figure 4 cells-10-01991-f004:**
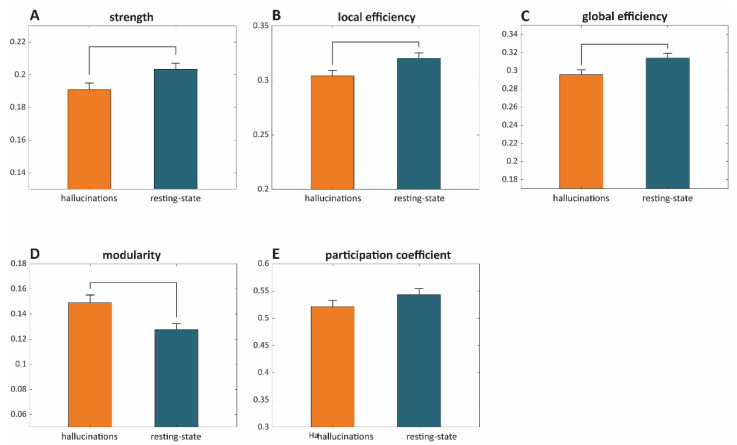
For each graph metric in the theta band, descriptive statistics for the two conditions are presented (mean + standard error). In each subplot, the orange bar represents the considered metric estimated during the VH condition, while the blue bar represents the metric related to the RS. Significant between-condition differences (*p* < 0.05 after FDR correction) are highlighted by a black line connecting the two bars. Graph strength is presented in (**A**), local efficiency in (**B**), global efficiency in (**C**), modularity in (**D**) and participation coefficient in (**E**).

**Figure 5 cells-10-01991-f005:**
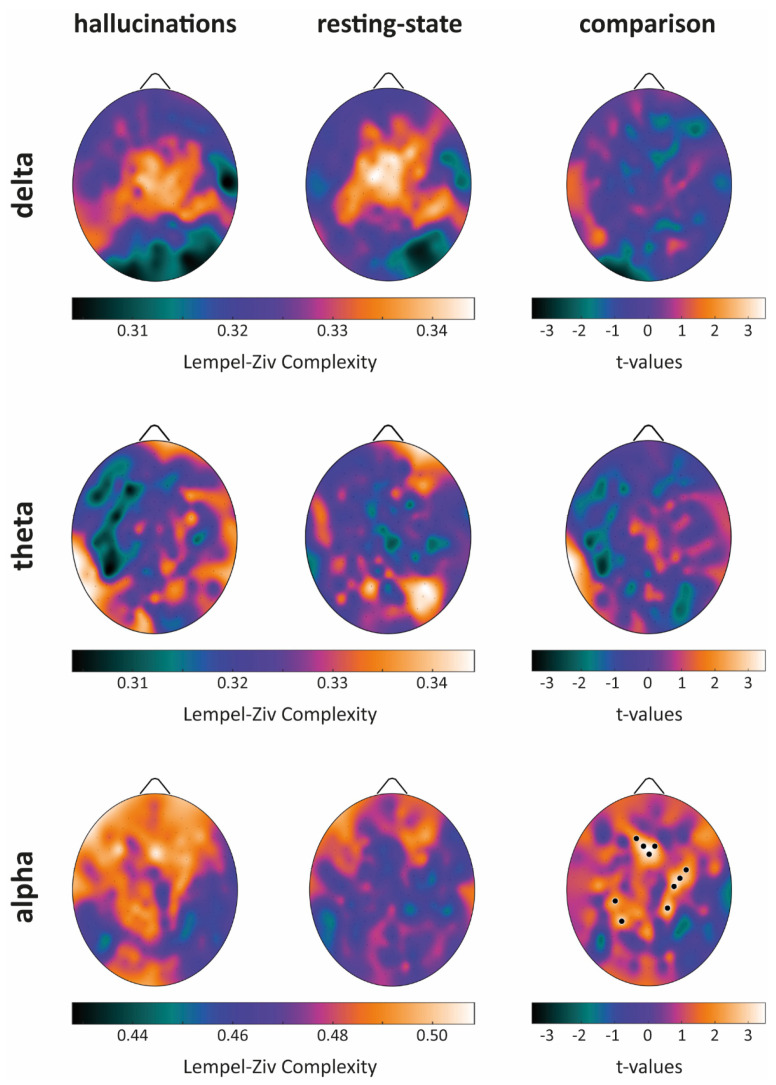
Average Lempel-Ziv complexity (LZC) scalp maps for each band and condition are presented in the first two columns. Band-wise comparisons are presented in the third column. Black dots indicate electrodes showing a significantly higher LZC during the VH condition as compared to the RS one, white dots (if present) indicate the opposite relationship. The critical t-value (in absolute value) for significance at *p* = 0.05 is |t| = 2.41 (single threshold tests for the maximum t-statistics).

## Data Availability

Some or all data used during the study are available from the corresponding author by request.

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
