# Peer review of "High-Density EEG in a Charles Bonnet Syndrome Patient during and without Visual Hallucinations: A Case-Report Study"

_cells, 2021, doi:10.3390/cells10081991_

Round 1
Reviewer 1 Report
Charles Bonnet syndrome (CBS) is a rare disease characterized by the occurrence of complex visual hallucination in elderly patients with severe or total loss of vision. So far, the neurobiological mechanisms underlying the hallucination remain elusive. Deafferentation of neurons in regions of visual cortex results in local neuronal hyperexcitability is one of the plausible mechanisms. In this case study, the authors used power density electroencephalogram (EEG) to investigate electrical activity changes during visual hallucination in a CBS patient. The authors found a significant reduction of power in frontal area, paralleled by an increase in midline posterior regions during visual hallucination. They concluded that hallucinations may stem from changes in visual cortex and otter core cortical regions. This study might shed some light into the pathophysiology of CBS.
Comments to the authors:
- Introduction: although the diagnosis of CBS in this patient is straightforward, did the authors exclude other rare causes of visual hallucination?
- Case report: the patient had positive family history of CBS, why? Genetic?
- EEG procedure: there had a 20-min of vivid visual hallucination during a 60-mi recording, why? Did this provoked? What’s the attack frequency of the patient?
- Methodology: no
- Discussion: no
Author Response
Reviewer #1
Charles Bonnet syndrome (CBS) is a rare disease characterized by the occurrence of complex visual hallucination in elderly patients with severe or total loss of vision. So far, the neurobiological mechanisms underlying the hallucination remain elusive. Deafferentation of neurons in regions of visual cortex results in local neuronal hyperexcitability is one of the plausible mechanisms. In this case study, the authors used power density electroencephalogram (EEG) to investigate electrical activity changes during visual hallucination in a CBS patient. The authors found a significant reduction of power in frontal area, paralleled by an increase in midline posterior regions during visual hallucination. They concluded that hallucinations may stem from changes in visual cortex and otter core cortical regions. This study might shed some light into the pathophysiology of CBS.
Comments to the authors:
- Introduction: although the diagnosis of CBS in this patient is straightforward, did the authors exclude other rare causes of visual hallucination?
We thank the reviewer for this question. Yes, the patient underwent a complete neurological examination by a certified neurologist to screen for the proper diagnostic.
- Case report: the patient had positive family history of CBS, why? Genetic?
We thank the reviewer for this relevant question. Unfortunately, we don’t have additional information regarding the facility history. To the best of our knowledge, no study has found a genetic factor contributing to the syndrome; but we think that further studies should investigate this issue.
- EEG procedure: there had a 20-min of vivid visual hallucination during a 60-mi recording, why? Did this provoked? What’s the attack frequency of the patient?
No, it was not provoked. As already written in the manuscript, the patient is not able to consciously control the occurrence of the visual hallucinations. Consequently, we consider ourselves lucky to have observed hallucinations during the EEG recording. At the time of the experiment, the patient reported experienced hallucinations serveral times a day (as already described in the manuscript).
Reviewer 2 Report
In their case report, Piarulli and colleagues aimed to characterize the cortical activity of a patient affected by Charles Bonnet syndrome, during visual hallucinations and during resting state. This case report aimed to serve as a pioneer for the technique used; the researchers focus on analysing the electrical activity of the brain with an EEG, instead of using MRI like in previous researches. The results were clearly written and interesting, opening many possibilities of future research regarding CBS. Specifically, the authors make a clear characterization of cortical activity correlated to visual hallucinations. The model proposed by the author supports the idea hallucinations could arise from widespread impairments in attention and/or perception, and stands on evidence from resting-state studies showing impaired connectivity within the Default Mode Network (DMN) and between brain attentional networks.
Although the manuscript hypothesis is very interesting, in my opinion the experimental methodology is weak and the results are not sufficiently clear to support the conclusions of the authors. I have several suggestions that the authors may want to take into consideration.
- Page 5, line 204 the word should be “Neuropsychological”
- Line 28 of the article we read that Charles Bonnet syndrome is described as present might lead to a misdiagnosis of some younger subjects
- The fact that this is a case study should be clear from the abstract to give the data analyzed the right consideration
- Since the study proposes a new model for a rare yet important syndrome, data from at least one control group and a late onset blindness control such should be fundamental to make any possible conclusion. to make any possible conclusion.
- In figure 2. The Cz electrode should be added or its absence explained.
- Could the authors report the whole topographic map of PSD for VH and RS conditions?
- In figure 5, although the LZC scalp maps for delta and theta bands were not significant, the authors should report them in the manuscript.
- While reading the Results section the authors mention subdivisions of alpha and theta bands into high and low range for the first time without clarifying and again later in the Discussion section which could confuse the reader. What was the rationale to use sub-band without a priori hypothesis?
- At this regard, how the starting hypothesis about the cortical hyperexcitability observed in the visual cortex of CBS patients is related to the main findings reported in the study? Could the authors discuss more deeply this point?
- The Conclusions section could be expanded with a clearer explanation about the neurophysiological mechanisms underlying the reported results and future implications of the findings rather than a proposal.
Round 2
Reviewer 2 Report
I would thank the authors for their responses.